# When scientific experts come to be media stars: An evolutionary model tested by analysing coronavirus media coverage across Italian newspapers

**Federico Neresini**[1]*, **Paolo Giardullo**[1], **Emanuele Di Buccio**[2], **Barbara Morsello**[1], **Alberto Cammozzo**[1], **Andrea Sciandra**[1], **Marco Boscolo**[3]

**1** Department of Philosophy, Sociology, Education and Applied Psychology, University of Padova, Padova, Italy, **2** Department of Information Engineering (DEI), University of Padova, Padova, Italy, **3** Department of Physics, University of Bologna, Bologna, Italy

* federico.neresini@unipd.it

**Data Availability Statement:** We made the "General Corpus A1" publicly available. We provided all information necessary for interested

## Abstract

The article aims to understand the process through which scientific experts gain and maintain remarkable media visibility. It has been analysed a corpus of 213,875 articles published by the eight most important Italian newspapers across the Covid-19 pandemic in 2020 and 2021. By exploring this process along the different phases of the management of the emergency in Italy, it was observed that some scientific experts achieve high media visibility—and sometimes notwithstanding their low academic reputation–thus becoming a sort of "media star". Scientific literature about the relationship between experts and media is considerable, nonetheless we found a lack of theoretical models able to analyse under which conditions experts are able to enter and to remain prominent in the media sphere. A Media Experts Evolutionary Model (MEEM) is proposed in order to analyze the main conditions under which experts can acquire visibility and how they can "survive" in media arena. We proceeded by analysing visibility of experts during SARS-CoV-2 pandemic and considering both their individual credentials previously acquired and the media environment processes of selection; MEEM acts hence as a combination of these two levels. Regarding the credentials, we accounted for i) institutional role/position, ii) previous media visibility, and iii) matches between scientific credentials and media competence. In our analysis, we collected evidence that high visibility in newspapers can be seen as evolutionary in the sense that some profiles—i.e. a particular configuration of credentials—are more adapt to specific media environments.

## 1. Introduction

This paper develops an analysis of experts' presence in the Italian quality press across the Covid-19 pandemic emergency in 2020 and 2021. Here we consider scientists and scientific advisors acting as experts, making use of their media over-exposure during the Sars-Cov-2

researchers to gain access to the data: URLs (the URLs where the article was published), sources (newspaper and feed/section where the article was published), datesPublished (dates when the article was published/updated). This dataset is adequate to replicate our study findings, and it has been deposited in a public repository, Zenodo, under the following doi: 10.5281/zenodo.7712714 (Data from: When scientific experts come to be media stars: an evolutionary model tested by analysing coronavirus media coverage across Italian newspapers).

**Funding:** The author(s) received no specific funding for this work.

**Competing interests:** The authors have declared that no competing interests exist.

emergency in order to investigate processes of the selection and enrolment of scientists to become media experts. We explored these processes across time along the different phases of the management of the emergency in Italy.

The literature about expertise and the media has reported how experts are recruited among scientists who are competent enough to say something relevant to a specific issue; however, one selection criterion is also the ready availability to be interviewed [1] and this feature does not necessarily ensure the quality of the person engaged as an expert [2, 3]. This already known process points to one of the paradoxes of the relationship between science and society as described by Bijker et al. [4]: "Scientific advice is asked for all kinds of problems . . . But as soon as advice is given, citizens, politicians, and NGOs comment on, criticize, or lend additional support to the scientists' report. The cases in which scientific advice is asked most urgently are those in which the authority of science is questioned most thoroughly" (p. 1). A deep and covert tension consists of the need for legitimacy by governments and scientific authorities as represented by public experts and non-scientists [5] who pose questions anyway about issues with technoscientific sounding background such as e.g. bioethics, environmental degradation and, especially in the current context, health; and moreover it is undeniable that media require experts as a resource for framing complex issues, such as precisely those related to health and the biosciences [6].

Media processes such as news production regimes have always been characterised by the need for rapid and effective communication to engage an audience that is assumed to be already overwhelmed by their daily life [3]. As Rachels succinctly put this: "reporters aren't interested in detailed analysis or lengthy qualifications. A short, pithy quote is what's wanted. Nor are reporters eager to hear reassurances that alarming events aren't alarming. That doesn't make good copy. What makes good copy is that the events being reported are morally troubling or worse" [7, p. 67]. These elements at the nexus between science communication and journalism studies are frequently noted in the literature [2, 8, 9].

All the contradictions we hinted at above become more evident in turbulent pandemic times when almost the entire daily life of billions of people has been reconfigured by governmental restrictions and recommendations provided by experts to cope with the spread of the pandemic. Moreover, experts were and still are enrolled by media to explain the development of the pandemic (e.g. data, therapeutic effectiveness, the importance of vaccination), both to reassure and to give an account of the efforts of citizens and governments to manage the uncertainty fostered by the virus.

There has perhaps never been a similar context in which scientists have been called to act as *public experts* in the media sphere with such urgency and so broadly. The pandemic therefore represents a great occasion for observing how "scientific expertise—transformed by the logic of mass media—enters the realm of policy-making" [p.132, 1] and that of the public sphere at the same time. Scientists who belong to a different context with norms and practices that might be far from media requirements are nonetheless urgently asked to adapt to specific communicative formats that follow different rules. Indeed, scientific credibility and the media requirement to be an effective public expert do not always meet [1, 10]; at the same time, being successful as a researcher (according to metrics such as the h-index) does not guarantee greater visibility or greater credibility.

Building upon this general background, our analysis addresses some recurring research questions about the *public experts* and their media role. More specifically, we explore those elements that contribute to the construction of profiles for those that emerge across the media as *public experts*. We will call those elements *credentials for expertise*. Therefore, we are interested in the relationship between media visibility and those credentials, considering media presence across the events that marked the development of the pandemic in Italy between 2020 and

2021. The pandemic context allows us to address questions about the relationship between media visibility and *credentials for expertise* as well as the processes that regulate the media visibility on which the role of *public expert* relies.

For this purpose, we propose an analytical model that we call the *Media Experts Evolutionary Model* (MEEM); the model is evolutionary in Darwinian terms, since it allows us to track the scientists that become resilient and thus adapt—thanks to credentials that are not necessarily scientific—to a media environment to which they do not originally belong. The MEEM can provide useful insights into the dynamic relationship between expertise and the media through metrics purposefully built to show who gains or loses visibility and which experts maintain their prominence across time within the media discourse. From there we can infer structural features about the process that brings scientists to become *media stars*, i.e. experts highly covered by media.

To give an account of the MEEM model, we will first offer a literature review about media and experts related to technoscientific and health issues, considering specifically media as a socio-technical environment that fosters or hinders the connection between the contents and the framing of the issues they discuss. Then we will describe how the model works in two phases: (1) the production of 'potential media experts' and (2) the 'media selection process', i.e., the process in which the media environment applies its own criteria for choosing experts. Our analysis develops across eight periods defined according to administrative interventions by Italian governments and concentrates on the top 25 scientists (i.e. most quoted) per period. We developed our analysis using the full coverage of Sars-Cov-2 by the top eight Italian quality press newspapers through the TIPS platform [11–13], additionally adopting an 'index of salience' that consists of the ratio of articles with scientifically related content to the total number of articles in the collection considered for the analysis [14]. What emerges clearly is that some features (pre-pandemic media visibility; gender; institutional role) contribute to keeping scientists visible as experts. Nonetheless, stability in the group of top 25 scientists does not ensure or correspond to scientific excellence, thus confirming some previous analysis in the literature. Some experts proved to be efficiently adaptative to the media environment which attempted to be reassuring, especially in most critical phases; in the Italian context, as already shown elsewhere [15], media reproduced the attempt to limit moral panic in support of governmental decisions on how to manage the pandemic. As we will emphasise in the conclusion, this reverberated strongly in the presence of those we label 'institutional experts' that better adapted to such a general condition.

## 2. Scientific experts and media

The issue of scientific experts in the media has been largely addressed within the debates on the Public Understanding of Science (PUS) and Public Communication of Science and Technology (PCST).

First and foremost, the constitutive dichotomy underlying the distinction between media and the scientific arenas [16] should be taken into consideration. Indeed, the so-called "two-arena model" [17] raises several questions, as it is not so easy to identify the boundaries between science and media.

In fact, media are concerned with science not only when they inform about research developments, according to the traditional trajectory of popularisation, but also when scientific knowledge is related to news events or to political, economic, cultural or sports issues [18]. Moreover, scientists find themselves involved in public controversies concerning several issues, as has happened in the past about nuclear energy [19], biotechnology [20, 21], and climate change [22, 23]—to name just a few. Therefore, taking the example of "scientific

controversies [. . .] covered by the media, scientific experts publicly disagree about facts and conclusions relevant for decision making, scientific misconduct leading to errors and deceptions is reported, and the interdependencies between science and industry or between science and politics are discussed as possible causes of bias in scientific knowledge" [16, p. 894]. However, the circulation of scientific expert discourses in media landscapes, especially regarding science, technology, and biomedicine, has shown how strongly they overlap with the policy discourses, revealing an underlying "elite sociotechnical imaginary" that has been described as "Science to the Rescue" [20] whereby science is framed in the public sphere as a solver of social problems and a driver of the economy. As pointed out by Petersen [24], in fact "mass mediated scientific expertise [is] either explicitly or implicitly linked to policy issues in the media coverage" (p. 868), often in contrast to formal scientific policy advice. In any case "scientific expertise in the mass media is observed by political decision makers and effectively enters the policy making process" [24, p. 869]. The mediatisation of science [25] to address controversial issues is moreover supported by the so-called "balance norm", i.e. the requirement that media has to ensure a balance of opposed views about a relevant issue, even if it doesn't correspond to the actual proportion of the pro/con fronts that are facing each other [26, 27].

Therefore, the mediatisation of science is imbricated with two further processes. First, the media are increasingly interested in reporting scientific contents, while scientists actively seek media attention [28, 29], at least because "most scientists consider visibility in the media important and responding to journalists a professional duty—an attitude that is reinforced by universities and other science organizations" [17, p. 14102]. For instance, academic researchers are increasingly encouraged to engage with the public [5, 30]. Second, scientists consequently try to exploit the role of "visible scientists" [31] since media coverage of researchers and scientific institutions may foster the visibility of scientific publications and thus their citations as well as the probability of receiving funding [32, 33].

It is important to note that these processes connected to media visibility for experts are "limited to a small number of scientists who appear in the media: it is only a small elite of reputable, media-savvy scientists who get the most prominence in the media" [25, p. 259].

The fact that only a few scientists become reputable, acquiring great visibility in the media arena, can be better understood by considering two specific pieces of evidence: on the one hand, "scientists' success in their field is a poor predictor of being cited by media, as the press tends to favour 'science celebrities' over specialists" [34, p. 2224]. The result is that a good scientist does not necessarily receive more attention from the media than a mediocre one, while a 'visible scientist' does not always hold a great reputation within the scientific community. Hence, media channels and scientific communities do not value reputation through the same parameters. On the other hand, most scientists are neither interested in entering the media context nor prepared to, simply because media visibility is irrelevant or not sufficiently relevant to gaining a reputation in the scientific community, where other reputational mechanisms operate. We can recognise that there is a cleavage between science and the media that is far from negligible. Although according to some we live in a mediatised society, where media work as an independent institution as well as an integrated part of other institutions [35], science and the media overlap only occasionally. This difference rebounds in the process of construction of the role of public expert: the scientist and the expert are two figures that do not automatically coincide, because the credentials collected by the former do not correspond to those required by the latter role [1].

The role of public expert, therefore, is not automatically acquired through studies and research work: as Martin aptly remarked, "expertness is an ascribed quality, a badge, which cannot be manufactured and affected by an expert himself, but rather only can be received from another, a client" [36, p. 159]. In this regard, the qualification of expert is attributed,

recognised and legitimated by others [37, 38]. Indeed, when scientists act as public experts, they are asked to respond to questions that move beyond the limits of their competence, dealing with issues they have not chosen [39]. Expertise then depends on the ability to propose solutions, to indicate practical ways out in the face of situations of uncertainty if not panic [40–42]. The Science and Technology Studies (STS) literature about experts, in fact, has devoted the most of its attention to the role of experts within the decision-making processes concerning public issues and/or policies. In such a perspective, experts mediate between the production of knowledge and its application; they define and interpret situations; and they set priorities for action in the public sphere [41].

Therefore, the media visibility of scientists become fundamental, and depends on two factors: 1) their adherence to media selection criteria, and 2) the recognition of their ability to act on the public stage as experts.

It is worth noting that what has been described so far is the result of activities in the science communication domain that took place under normal conditions. Similar processes are also at work when routines are reconfigured by exceptional events, such as in the case of a pandemic crisis; their action is indeed enormously emphasised by the conditions of the ongoing crisis, requiring explanations, urgent decisions, reassurance that the situation is under control, foresight about what will happen both in the short- and in the mid-term [43]. For this reason, the pandemic provides an interesting opportunity to make such processes more visible and thus to deepen the analysis of the role of scientists acting as public experts. The tension between "attention of" the media and "attention to" the media [29], for example, has become even more evident during the first two years of pandemic, even if the spasmodic search for information by the media—both to cover the issue of the day (actually a day prolonged for a couple of years, at least) and to provide information required by their audiences—could seem to push into the background the active seeking of media attention by the scientists. But this seems to not be the case: scientists too have actively taken part in the media coverage about the pandemic.

It is important also to consider whether the mediatic presence of scientists changes over time, and whether some among them can profit from a kind of 'positioning revenue'—i.e. whether the visible scientists tend to be always the same. In light of these considerations, two research questions can be outlined:

RQ1: Who is taking the stage as a scientific expert in the media discourse about Covid19?

RQ2: How does the prominence of various scientific experts change during the Covid-19 pandemic?

These RQs allow us a more in-depth examination of 1) the presence of scientific experts in the media arena during the Covid-19 pandemic, and 2) how these changes over time—with the aim of analysing the process through which experts enter the media arena and become "visible scientists". As in the legal context [44–47], in fact, also in the mediatic context experts are co-produced by the interaction of many actors that follow different criteria for deciding both who are the most appropriated experts to speak about the Covid-19 pandemic, and which credentials are more suitable to gain a prominent position and become trustable by the public. Media, as socio-technical environments, thus have the power of a) configuring the frame in which relevant issues are treated and b) assigning experts authority even in areas outside their professional competences [48–50] lending credibility to maverick scientists [51]. At the same time, it has to be considered that expertise and credibility are associated with the communicators' credentials, which are a mix of institutional affiliations, leadership positions, and academic recognition [52]; moreover, this results in the public becoming more influenced by visibility than expertise when the experts' credibility must be evaluated [34]. Given these premises, and with the aim of addressing the RQs, we propose an evolutionary model for analysing

how experts can gain access to the media scene. According to a Darwinian approach, if scientific competence, h-index, and scientific reputation in general are not enough to reach media visibility, we can imagine a sort of selection process set up by media as socio-technical environments in which specific criteria are relevant to being selected and thus to becoming a *media star*. We may postulate that the experts who best fit those criteria have a chance to become a *media star* and thus to remain in the arena of public debate for a long time. We can also assume that the conditions set by the Covid-19 pandemic provide the opportunity to better analyse the processes we are interested in.

The proposed MEEM model envisages two phases: the first concerns the recognition of 'potential media experts' based on relevant socio-academic and/or institutional credentials; the second encompasses the actual selection process of media experts by applying media-sphere criteria.

Hence, becoming 'potential media experts' requires first acquiring some credentials depending on specific conditions that are socially recognised as relevant in each context. Such conditions can be very general—i.e. shared by a large social context; others operate within narrower situations, such as for example the institutional structure of a country or rules that are in effect during a given time span. In our case, i.e. considering Italy over the Covid-19 outbreak, the most relevant are:

- credentials achieved by the completion of an academic training course (academic expert—AE);

- credentials deriving from institutional positions, i.e. roles within the public administration or within a public consultant organism, in both cases as a result of routine processes (ordinary institutional expert—OIE);

- credentials depending on institutional positions during a crisis, for example a pandemic [53]—as part of an *ad hoc* techno-scientific advisory board or through appointment as special commissioner for coping with a particularly problematic situation (emergency institutional expert—EIE);

- credentials acquired on the base of personal experience, i.e. patients and their relatives or other persons with a close relationship, like that of relatives, friends and volunteers (personal experience expert—PEE);

- credentials that result from a mix of academic titles and experience in the field, i.e. physicians or other health workers (academic and personal experience expert—APE) or academics with a leading role in business activities such as CEO (AB).

The second phase can be described as a *media sieve*. The media act as a socio-technical environment in which potential experts are selected. In other words, potential experts become *media stars* only if they are suitable for the media context.

In this phase as well, the selection criteria can be identified at a very general level, but obviously they are specified differently depending on contexts:

- accessibility, i.e., "whether scientists reply to emails and messages, how quickly they reply, and how 'complicated' they are in the interactions" [1, pp. 137–138];

- previous visibility in the media arena;

- being in the contact list of journalists or other communication professionals (contactability);

- capability to promote themselves as useful and/or interesting to media operators;

- editorial suitability, i.e., experts that sound in line with the media channel orientation [1, 54] or that balance opposite positions toward the issue at stake [55, 56];

- media reputation, i.e. "using academics as sources tends to increase the credibility of the news and provide 'unbiased' viewpoints" [57, p. 87];

- media appropriateness, i.e. being suitable to the operative rules and rhythms of media (ability to communicate, be concise, *physique du role* and so on).

Media experts, and especially the most visible among them, are seen as able to offer solutions to questions that are deemed very relevant by the public, responding at both a pragmatic level (what should we do?) and a sense-making level (why is this happening?). At the same time, they respond to mediatic selection criteria. In addition, media experts are often not producers of first-hand knowledge; rather, they act as mediators "between the knowledge producers and the knowledge users" [58, p. 186], thus participating in the process of knowledge production as also emphasised by Latour [59].

Media experts, in fact, often contribute to the flow of media communication, lending themselves to intervene even on subjects and in areas that are not within their competence [51]. Thus, in the Italian media context, we have seen distinguished scientists providing their expertise on issues from health policies to the Russia-Ukraine crisis, from distance learning issues to the Sanremo music festival. This happens also because each newspaper like each television network has its own *trusted experts*, available to give their advice about many different issues. Media experts thus could be regarded at first glance as intellectuals [58, p. 177; 60, pp. 19–20], while the fact that their expertise represents a point of view embodied in specific media channels make them closer to the figure of the "organic intellectual" or the "maverick scientist" [51]. This can be regarded as a specific part of the issues related to RQ1.

Consequently, an evolutionary approach to the media selection process of scientific experts seems to suggest, on the one hand, a lesser relevance than that which the social sciences usually ascribe to the role of experts in the knowledge society [42, 58, 61]; but, on the other hand, it provides a useful point of view to enrich such a debate by offering further insights deriving specifically from a sharpened awareness of the strategic role of the media in the context of the knowledge society, not only as a channel for the dissemination of knowledge, but also as an arena within which knowledge is shaped and made socially available, precisely by the experts. Therefore, although the "democratization of expertise" [58, p. 171] finds its full application in mediascapes, the proposed evolutionary model highlights that there is still an elite of experts, selected by the media.

## 3. Methodology

To answer our RQs we have analysed the coverage of the Covid-19 outbreak from 01.01.2020 until 31.12.2021 (General Corpus A1 = 572,944 articles), taking into consideration the online versione of the eight most important Italian newspapers (namely il Corriere della Sera, la Repubblica, la Stampa, il Sole24Ore, il Giornale, Avvenire, il Messaggero, il Mattino). The 24 months' timespan has been divided into periods according to the sequence of governmental decrees stating rules to contain the contagion and then outlining steps toward a relief of restrictions. In Table 1 we report the corpus composition for articles referring to Covid-19 (N = 213,785) selected through the query 'covid' OR 'corona virus' OR 'coronavirus' through TIPS platform; we called it 'Covid General Corpus' (A2). From there we extracted two sub-corpora of articles containing at least one expert with scientific credentials (N = 25,550): one is the 'General Expert Subset' (B1) to be contrasted with 'Covid Expert Sub-Set' (B2). The scientists cited by the articles have been identified through a specifically developed Named Entities

**Table 1. Corpora composition and size, in the 24 months' time span, January 1st, 2020 –December 31st, 2021.**

| | CORPUS (A) | | | CORPUS (B) | | |
|---|---|---|---|---|---|---|
| | GENERAL CORPUS (total published articles) A1 | COVID GENERAL CORPUS A2 | % Covid general corpus on general corpus | GENERAL EXPERT SUBSET (art citing at least 1 scientist within GENERAL CORPUS) B1 | COVID EXPERT SUBSET (art. citing at least 1 scientist within COVID GENERAL CORPUS) B2 | % Covid expert subset of general expert subset |
| total published articles | 572,944 | 213,785 | 37.31 | 33,841 | 25,550 | 75.50 |
| articles related to technoscience | 50,608 | 25,895 | 51.17 | 15,778 | 11,940 | 75.67 |
| technoscientific salience (% of technoscience-related articles within the corpus) | 8.83 | 12.11 | | 46.62 | 46.73 | |

Recognition (NER) procedure that first detected candidate entities—i.e. personal names—and then selected those that are listed in Web of Science, Scopus and PubMed. This screening process returned 774 single entities; the number of articles mentioning each of them has been counted for all the eight most important Italian newspapers both in the General Corpus (A1) and in the Covid General Corpus (A2). We have thus obtained the number of articles in which each scientist has been cited in each newspaper, both in articles that refer to Covid-19 and those that don't. Shares are calculated as the ratio of articles mentioning each scientist to the total number of articles mentioning all scientists, representing how each newspaper has divided its mentions between the 774 most mentioned scientists. Subsequently, we have calculated the 'scientists' share', i.e. the ranking showing the most visible scientists, in order to obtain a measure of the prominence of experts in the media.

For the current analysis we considered the top 25 scientists in each corpus, not only to stay focused on the most visible one (*media star*), but also because the 25 top scientists cover 50% of the articles in which at least one scientist is quoted. Looking at the complete scientists' list, it is easy to identify a group of "evergreen" names, i.e. Albert Einstein, Charles Darwin or Galileo. They have been removed whenever they appeared in the top 25 most visible scientists, as the purpose of this research is to bring out those scientists most closely linked to current topics and the related controversies, i.e. where they play the role of public experts.

With the aim of addressing RQ1, we have also analysed the relationship between media visibility (i.e. share) and scientific production (i.e. h-index), considering that the last can be interpreted as *scientific visibility*.

RQ2 requires dividing the time span 2020–21 into periods (Table 2). For this purpose, we followed the sequence of administrative decrees introduced by the Italian government to tackle

**Table 2. Pandemic periods segmented according to measures (administrative decrees) successively introduced by the Italian government.**

| PERIOD | FROM | UNTIL | DESCRIPTION |
|---|---|---|---|
| 1 | 01.01.20 | 08.03.20 | first wave of infections |
| 2 | 09.03.20 | 04.05.20 | lockdown |
| 3 | 05.05.20 | 07.10.20 | progressive reduction of restrictive measures |
| 4 | 08.10.20 | 03.11.29 | second wave of infections |
| 5 | 04.11.20 | 12.02.21 | new restrictions |
| 6 | 13.02.21 | 16.06.21 | third wave of infections |
| 7 | 17.06.21 | 14.10.21 | Green Pass (not mandatory) |
| 8 | 15.10.21 | 05.12.21 | mandatory Green Pass |
| 9 | 06.12.21 | 31.12.21 | mandatory Super Green Pass (3rd dose vaccine) |

the pandemic with sets of measures and behavioural rules (see the Italian Ministry of Health website https://www.salute.gov.it/portale/nuovocoronavirus/archivioNormativaNuovoCoronavirus.jsp). The nine periods thus obtained are the following:

The most visible experts in the media arena were then classified by considering the credentials through which they derive "expertness" [36]. This made it possible to detect which types of experts become *media experts* and therefore to observe how the *media sieve* works.

## 4. Results

The articles mentioning Covid-19 (COVID GENERAL CORPUS A2) are 37.31% of all articles published by the eight TIPS-monitored newspapers in the timespan ranging from 01.01.2020 to 31.12.2021 (GENERAL CORPUS A1). As already shown in another analysis of Italian daily quality press coverage [15], pandemic is a major driver for technoscientific salience, i.e. the ratio of articles related to technoscience to the total of published articles. The articles about Covid-19 are indeed characterised by a more prominent technoscientific frame, as the salience rises to 12.11% compared to the salience of total articles (8.83%) (see Table 1A). The Covid-19 contribution to technoscientific salience becomes more evident if we consider the share of Covid-19-related articles among the articles dealing with technoscientific issues, which comes to 51.17%. This makes Covid-19 the main issue of articles dealing with technoscience. In applying the same kind of analysis for the subset of articles that name at least one expert (Table 1B), we noticed that there is no significant increase in salience or in the contribution to the number of articles about science and technology. Interestingly, we can also note that the presence of experts is almost equally shared among articles related to Covid-19 and those with technoscientific relevant content.

For the current analysis we opted to focus specifically on B1 and B2 corpora, which are populated by articles in which at least one scientist is mentioned. On this basis, we will first tackle RQ1 and then we will move to RQ2.

### 4.1. Who is taking the stage as a scientific expert in media discourse about Covid-19?

To properly discuss RQ1, we will unpack it as three main issues. The first concerns the credentials that supports the prominence of experts: data analysis allowed us to obtain a list of the most prominent experts on the media scene in relation to the Covid-19 topic and, therefore, to extrapolate which credentials were the most relevant for making oneself suitable to the media environment and thus to the public discourse about the pandemic. Second, to better understand the processes driving those experts to the frontstage during the Covid-19 pandemic, we compared each expert presence during the pandemic emergency with the pre-pandemic visibility. Third, in order to check whether the academic credentials may play a role, we compared the media visibility of each expert with their *scientific visibility* (h-index) in order to verify the extent to which scientific reputation works as a predictor of media prominence.

Among the top 25 scientists most prominent in the media during the first two years of the pandemic, three main categories predominated in articles speaking generally about Covid-19 (Table 3): institutional expert, either ordinary (OIE) and/or emergency (EIE); academic expert (AE); and academic with personal engagement with the disease due to his or her clinical experience (APE).

Nine of the experts with the highest 'share' are institutional ones (OIE and or EIE), meaning that the most visible credentials rendering them suitable for the media environment derive from institutional positions occupied before the pandemic or because of the pandemic. Six of them, in fact, have become members of the Italian Scientific-Technical Committee–i.e. the

**Table 3. Top 25 media star experts in all Covid-related articles in the 24 months' time span, January 1st, 2020 –December 31st, 2021.**

| COVID EXPERT SUBSET (25,550 articles, i.e. all articles mentioning COVID-19 in which at least 1 scientist is cited) | | | | COVID EXPERT SUBSET (only articles related to technoscience among those mentioning COVID-19 and in which at least 1 scientist is cited—i.e. 11,940 articles) | | | |
|---|---|---|---|---|---|---|---|
| | expert category | num. articles | share | | expert category | num. articles | share |
| Silvio Brusaferro | OIE & EIE | 1031 | 4.04 | Fabrizio Pregliasco | APE | 339 | 2.84 |
| Walter Ricciardi | OIE | 936 | 3.66 | Roberto Burioni | AE | 335 | 2.81 |
| Anthony Fauci | OIE | 921 | 3.60 | Gianni Rezza | OIE & EIE | 333 | 2.79 |
| Gianni Rezza | OIE & EIE | 908 | 3.55 | Anthony Fauci | OIE | 316 | 2.65 |
| Franco Locatelli | OIE & EIE | 778 | 3.05 | Silvio Brusaferro | OIE & EIE | 293 | 2.45 |
| Roberto Burioni | AE | 772 | 3.02 | Andrea Crisanti | AE | 255 | 2.14 |
| Fabrizio Pregliasco | APE | 770 | 3.01 | Massimo Galli | APE | 250 | 2.09 |
| Massimo Galli | APE | 738 | 2.89 | Walter Ricciardi | OIE | 241 | 2.02 |
| Andrea Crisanti | AE | 733 | 2.87 | Matteo Bassetti | APE | 217 | 1.82 |
| Matteo Bassetti | APE | 607 | 2.38 | Franco Locatelli | OIE & EIE | 206 | 1.73 |
| Nino Cartabellotta | AE | 522 | 2.04 | Ilaria Capua | AE | 200 | 1.68 |
| Pier Luigi Lopalco | AE | 454 | 1.78 | Roberto Cauda | APE | 199 | 1.67 |
| Ilaria Capua | AE | 421 | 1.65 | Massimo Ciccozzi | AE | 160 | 1.34 |
| Alberto Zangrillo | APE | 346 | 1.35 | Pier Luigi Lopalco | AE | 154 | 1.29 |
| Agostino Miozzo | EIE | 340 | 1.33 | Giorgio Palù | OIE & EIE | 151 | 1.26 |
| Francesco Vaia | AE-HOD | 303 | 1.19 | Massimo Andreoni | AE | 141 | 1.18 |
| Enrico Giovannini | MIN | 302 | 1.18 | Antonella Viola | APE | 137 | 1.15 |
| Roberto Cauda | APE | 277 | 1.08 | Alberto Zangrillo | APE | 137 | 1.15 |
| Giorgio Palù | OIE & EIE | 276 | 1.08 | Nino Cartabellotta | AE | 116 | 0.97 |
| Massimo Andreoni | AE | 242 | 0.95 | Alberto Mantovani | AE | 114 | 0.95 |
| Giuseppe Ippolito | AE & EIE | 240 | 0.94 | Sergio Abrignani | EIE & AE | 113 | 0.95 |
| Antonella Viola | APE | 229 | 0.90 | Giuseppe Ippolito | AE & EIE | 111 | 0.93 |
| Nicola Magrini | OIE & EIE | 208 | 0.81 | Nicola Magrini | OIE & EIE | 91 | 0.76 |
| Roberto Cingolani | MIN | 203 | 0.79 | Giordano Beretta | APE-IST | 91 | 0.76 |
| Massimo Ciccozzi | AE | 195 | 0.76 | Guido Silvestri | AEM | 88 | 0.74 |
| | | 12752 | 49.91 | | | 4788 | 40.10 |

board of scientific experts appointed by the Italian government—thus gaining further media visibility from their institutional positions related to the Covid-19 outbreak. Moreover, institutional experts have the highest share. Hence the combination of OIE and EIE credentials—which we can also ascribe to the two ministers, Giovannini and Cingolani, who also have academic credentials—increases the likelihood of being suitable for the media environment. In the specific, Cingolani is a physicist and academic; before being appointed minister of Ecological Transition in 2021 he was scientific director of the Italian Institute of Technology. Giovannini is an academic—an economist by training and an expert in statistics—who served twice as minister and led the Italian National Institute of Statistics.

As an example, Silvio Brusaferro—a physician who ranks first among the experts most prominent in the public discourse on Covid-19 and is the spokesperson for the Scientific-Technical Committee—alone represents a share of almost one tenth within the B2 corpus.

Other institutional experts taking a prominent position in the press are Walter Ricciardi and Anthony Fauci, the head of the Istituto Superiore di Sanità (the Italian Health Institute) and the director at the NIH (National Institute of Allergy and Infectious Disease in the US), respectively. Fauci, although not part of the institutions directly involved in managing the emergency in Italy, has enjoyed a notable presence in the Italian media by virtue of his

institutional role. We can also include Francesco Vaia, the Director of the Italian Center for Infectious Diseases, among the institutional experts.

The other two experts' categories represented within the *media stars* are academic experts (AE) and clinicians (APE). This means that also academic credentials and/or a specific expertise acquired through a direct contact with disease can provide access to the media forefront, although their relevance obviously depends on the issues addressed and the socio-cultural context [57]. During the hot crisis of early Covid-19 pandemic, undoubtedly academic experts played a significant role, although the political discourse took over almost from the beginning [15].

It is worth noting, against this general background, that gender represents a strong predictor of media visibility: only Ilaria Capua and Antonella Viola appear in the top 25 ranking. Specifically, Ilaria Capua is a virologist with great achievements in the study of avian influenza and currently director of the One Health Center of Excellence at the University of Florida. Antonella Viola is an immunologist professor of general pathology at University of Padua. Although this finding aligns with other studies about the gender gap in the presence of researchers in the media, several other factors can explain or influence such a gap, including seniority or age, which are often connected to the structure of the academic community [62]. Therefore, it is unclear if such a low performance in media visibility is the effect of a proper media process or the reflection of a more widespread gender gap. What seems clear is that, at least in this case, female gender cannot be regarded as a positive factor directly affecting the credentials for experts' media suitability.

Similar results emerge when we look to the top 25 media experts only considering articles related to both Covid-19 and to technoscience (B2); nonetheless we recorded three interesting differences.

First, although institutional experts still predominate (10 out 25 experts), they are less represented within the highest positions of rank. In contrast, it is possible to find at the top of the ranking more clinicians and academics. This suggests that they are particularly associated with discourses focusing on explaining the bio-medical reasons of the emergency, rather than suggesting policy solutions. This is the case, for example, of Fabio Pregliasco, who occupies the first place among the most cited experts, with a share of 7.06; he is a virologist and associate professor at the State University of Milan and medical director of the Galeazzi Hospital in Milan.

Secondly, women remain strongly under-represented, but they increase their share. Antonella Viola gained media visibility during the pandemic, and Ilaria Capua was already highly exposed during previous years, both for scientific reasons—she not only sequenced the genome of the SARS virus but decided to make the sequence freely accessible—and for being accused of having smuggled samples of the virus to pharmaceutical companies and being proven innocent in 2016.

Third, we recorded a category of academics who already had media visibility because of their engagement in public communication of science activities. Roberto Burioni falls under this category, since he represents one of those experts already accredited as media expert because of his engagement in the Italian controversy about compulsory vaccination for children in 2017. His mediatic relevance is demonstrated by the fact that the neologism *burionism* has become a common term to describe opinionated criticism of anti-vaxxers. During the Covid-19 pandemic Roberto Burioni became even more central, climbing the ranking by four positions to become the second most present academic expert (AE) in the mediascape.

In general, being already present in the media appears to be an important factor affecting the suitability to the media environment. More than a quarter of top 25 experts during

**Table 4. Top 25 experts' media prominence before the pandemic (2010–2019).**

| Pre-pandemic (2010–2019) media visibility of the top 25 experts during the pandemic [*] | | | Visibility during the pandemic (2020–2021) [share] | |
|---|---|---|---|---|
| | number of articles mentioning the expert | expert category during the pre-pandemic period | | Share |
| Enrico Giovannini | 1184 | AE & MIN | Silvio Brusaferro | 4.04 |
| Walter Ricciardi | 394 | OIE | Walter Ricciardi | 3.66 |
| Alberto Zangrillo | 247 | APE | Anthony Fauci | 3.60 |
| Roberto Burioni | 227 | AE | Gianni Rezza | 3.55 |
| Roberto Cingolani | 181 | AE & OIE | Franco Locatelli | 3.05 |
| Fabrizio Pregliasco | 174 | APE | Roberto Burioni | 3.02 |
| Ilaria Capua | 131 | AE | Fabrizio Pregliasco | 3.01 |
| Giuseppe Ippolito | 126 | AE & EIE | Massimo Galli | 2.89 |
| Gianni Rezza | 113 | AE & EIE | Andrea Crisanti | 2.87 |
| Massimo Galli | 84 | APE | Matteo Bassetti | 2.38 |
| Franco Locatelli | 83 | OIE | Nino Cartabellotta | 2.04 |
| Anthony Fauci | 60 | OIE | Pier Luigi Lopalco | 1.78 |
| Massimo Andreoni | 46 | AE | Ilaria Capua | 1.65 |
| Nino Cartabellotta | 32 | AE | Alberto Zangrillo | 1.35 |
| Matteo Bassetti | 18 | APE | Agostino Miozzo | 1.33 |
| Pier Luigi Lopalco | 17 | AE | Francesco Vaia | 1.19 |
| Agostino Miozzo | 16 | EIE | Enrico Giovannini | 1.18 |
| Giorgio Palù | 16 | OIE & EIE | Roberto Cauda | 1.08 |
| Silvio Brusaferro | 14 | OIE & EIE | Giorgio Palù | 1.08 |
| Roberto Cauda | 9 | APE | Massimo Andreoni | 0.95 |
| Andrea Crisanti | 8 | AE | Giuseppe Ippolito | 0.94 |
| Antonella Viola | 4 | APE | Antonella Viola | 0.90 |
| Francesco Vaia | 3 | APE-DS | Nicola Magrini | 0.81 |
| Nicola Magrini | 3 | OIE | Roberto Cingolani | 0.79 |
| Massimo Ciccozzi | 0 | AE | Massimo Ciccozzi | 0.76 |

pandemic were in fact already highly media-visible in the ten years (2010–2019) before the pandemic (Table 4).

It is also interesting to note that Roberto Cingolani served as Minister starting in 2021, while Enrico Giovannini—who also was already serving as Minister by 2021—had also been Minister for one year between 2013 and 2014. Thus, both had a relevant role as institutional experts, the first as Director of the Italian Institute of Technology (IIT), the second as President of the Italian Institute of Statistics (ISTAT). This means that their media visibility can be ascribed only partially to their ministerial role, and mainly in the case of Giovannini.

Other experts gained a significative media visibility before the pandemic for different reasons—for example, Zangrillo as the personal physician of former prime minister Silvio Berlusconi, Burioni because of the vaccine controversy, and Capua for her supposed implication in a legal issue. This confirms that being in the media previously is a driver for their media visibility during the pandemic. It should be further noticed that media visibility of these experts is not only antecedent, but it also entails transmediality, namely to be exposed across different types of media: Burioni, for example, has been not only often invited to TV talk shows [63], but he has been present also on social media and in general on the web (blogs or sites more or less focused on scientific topics). This highlights that there is a significant relationship between presence in newspapers and presence in other widely distributed media. Although our study concerns the daily quality press, what we can observe about the visibility of experts appears

**Table 5. Comparing scientific and media visibility of the top 25 experts across 24 months' time span, January 1st, 2020 –December 31st, 2021.**

| COVID EXPERT SUBSET (25,550 articles) | | | COVID EXPERT SUBSET (only articles related to technoscience—11,940 articles) | | |
|---|---|---|---|---|---|
| | **h-index** | **share** | | **h-index** | **Share** |
| Silvio Brusaferro | 24 | 4.04 | Fabrizio Pregliasco | 19 | 7.08 |
| Walter Ricciardi | 43 | 3.66 | Roberto Burioni | 26 | 7.00 |
| Anthony Fauci | 192 | 3.60 | Gianni Rezza | 11 | 6.95 |
| Gianni Rezza | 11 | 3.55 | Anthony Fauci | 192 | 6.60 |
| Franco Locatelli | 91 | 3.05 | Silvio Brusaferro | 24 | 6.12 |
| Roberto Burioni | 26 | 3.02 | Andrea Crisanti | 62 | 5.33 |
| Fabrizio Pregliasco | 19 | 3.01 | Massimo Galli | 55 | 5.22 |
| Massimo Galli | 55 | 2.89 | Walter Ricciardi | 43 | 5.03 |
| Andrea Crisanti | 62 | 2.87 | Matteo Bassetti | 61 | 4.53 |
| Matteo Bassetti | 61 | 2.38 | Franco Locatelli | 91 | 4.30 |
| Nino Cartabellotta | 1 | 2.04 | Ilaria Capua | 50 | 4.18 |
| Pier Luigi Lopalco | 34 | 1.78 | Roberto Cauda | 6 | 4.16 |
| Ilaria Capua | 50 | 1.65 | Massimo Ciccozzi | 39 | 3.34 |
| Alberto Zangrillo | 60 | 1.35 | Pier Luigi Lopalco | 34 | 3.22 |
| Agostino Miozzo | 0 | 1.33 | Giorgio Palù | 59 | 3.15 |
| Francesco Vaia | 8 | 1.19 | Massimo Andreoni | 39 | 2.94 |
| Enrico Giovannini | 5 | 1.18 | Antonella Viola | 40 | 2.86 |
| Roberto Cauda | 6 | 1.08 | Alberto Zangrillo | 60 | 2.86 |
| Giorgio Palù | 59 | 1.08 | Nino Cartabellotta | 1 | 2.42 |
| Massimo Andreoni | 39 | 0.95 | Alberto Mantovani | 180 | 2.38 |
| Giuseppe Ippolito | 65 | 0.94 | Sergio Abrignani | 57 | 2.36 |
| Antonella Viola | 40 | 0.90 | Giuseppe Ippolito | 65 | 2.32 |
| Nicola Magrini | 23 | 0.81 | Nicola Magrini | 23 | 1.90 |
| Roberto Cingolani | 84 | 0.79 | Giordano Beretta | 32 | 1.90 |
| Massimo Ciccozzi | 39 | 0.76 | Guido Silvestri | 104 | 1.84 |

thus a proxy of the whole media sphere, even if it needs to be deepened through further analysis.

Thus, MEEM allows us to observe that the *media sieve* relies mainly on three factors that affect the process of becoming a *media star* during the pandemic: 1) occupying an institutional position, 2) having previous media visibility (whatever the reason for this), and 3) displaying a combination of academic credentials with the media competence to act as an expert, which means offering explanations and practical suggestions deemed useful by the media and suitable to their modus operandi.

Regarding the last point, comparing the share and h-index of the top 25 experts allows us to better understand the relationship between media visibility (share) and *scientific reputation* (h-index), making their mismatch clear: we recorded a very low correspondence between scientific reputation and media visibility, as Table 5 shows. A consideration of the first 25 positions in share for the Corpus B1 and those for Corpus B2 of articles related to technoscience shows that the correlations are indeed not significant (R is respectively 0.06 and –0.17, while the p value is 0.78 and 0.42).

Thus, as already noted in the literature, scientists and experts do not automatically coincide [1, 64]. If being considered an expert is determined by several factors, including circumstances that do not have an immediate connection to scientific knowledge and expertise [65], this likewise pertains to visibility in the media. In this case there are at least three criteria that intervene

as selectors: i) the previous presence in the media, ii) the institutional role/position; and iii) the match between credentials and media competence.

## 4.2. How does the prominence of scientific experts vary across time during the Covid-19 pandemic? Continuities and discontinuities in pandemic times

In the previous paragraph we highlighted the characteristics that favour scientists becoming public experts across media. We noticed that most of articles reporting the name of at least one expert tend to refer to few scientists. The presence in the media of a (relatively) limited group of scientists is a well-documented phenomenon [25]. Acquiring media visibility involves scientists becoming included in feedback loops of media attention; these loops drive scientists to be consistently reported/interviewed in the media across time, shaping a positive effect whereby those already present in the media become more and more visible [1].

In late '70s, Goodell defined this feature as the "visible scientist" [31]. This concept captures the idea of visibility as a condition that can be acquired. As described in previous paragraphs, although scientists may try to enter the media arena for several strategic/personal reasons, the fact that only a limited number of them are able to maintain a presence in the media hints that the process is bi-directional; that is, once media select *suitable* experts among those with some scientific credentials, they *sieve* those better aligning with media criteria. As postulated by Goodell and further confirmed more recently, the selection of experts in the media tends to become stable [1, 25, 31]. As shown elsewhere [12], although the number of actors implicated in an issue may increase, still the largest share consists of those actors who came first, thus creating a group of subjects that dominates the issue. In a sense, first-comers occupying a specific issue in media environments tend to become 'dominant', further limiting the access of new actors.

Given these premises, addressing RQ2 offers two opportunities: first to determine whether in a pandemic context those kinds of process are confirmed; second, to explore more deeply how the process of experts' selection works. More specifically we can check if the firstcomer rule is challenged by other experts' features (e.g. credentials), thus hinting at an evolutionary model. In other words, through this analysis it will be possible to explore further whether media have an active role in sieving experts independently according to variables other than early presence in articles related to a specific issue.

To do so, we conducted an analysis of the top scientists across time and plotted this through heatmaps. Chronological heatmaps are based on the descending ranks of the top 25 experts in each period. Included in the figures are scientists with at least two appearances in the top 25 of the nine periods. To analyse the experts' appearances in the eight quality Italian newspapers, we computed each expert's share of mentions within each newspaper. The corresponding heatmap cells show the shares of each scientist within each newspaper. Shares are computed relative to the total number of mentions, unlike the chronological heatmaps, where the rankings only referred to the top 25 experts of each period (or a value of 0 was assigned if an expert does not appear in the top 25 in a given period). Scientists in all heatmaps in the text are sorted by category.

More specifically, we analysed the top scientists' permanence during the nine periods under examination here both in all articles related to Covid-19 (Corpus B1 –Fig 1) as well as in the corpus restricted to the articles related to technoscience (Corpus B2 –Fig 2).

The dominance of firstcomers appears plausible in this case as well: analysing the two heatmaps (Figs 1 and 2) shows clearly how those who were already present as key experts in P1 (the early phase of Sars-Cov-2 virus spreading in China, but before the declaration of

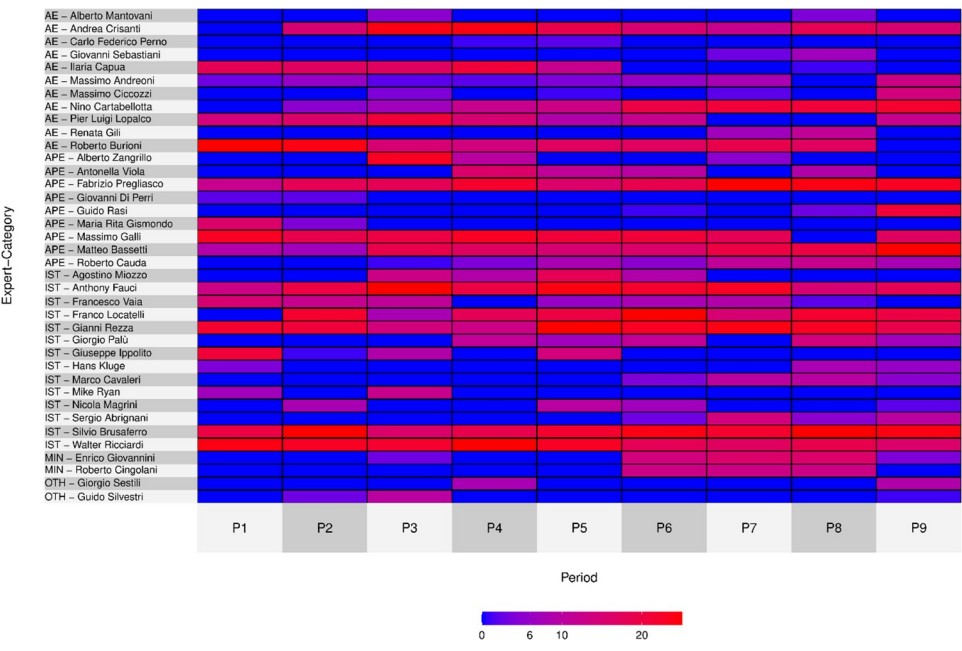

**Fig 1. Heatmap share for experts across time, 24 months' time span, January 1st, 2020 –December 31st, 2021 (B2 COVID EXPERT SUBSET, N = 25,550).**

emergency in Italy) tend to keep their prominence across time. This is the case for the AE Burioni, the APEs Pregliasco, Galli and Bassetti, and finally the ISTs (i.e. OIE and/or EIE) Brusaferro and Ricciardi; similarly, Anthony Fauci, nominated in January 2020 as chief for the U.S.

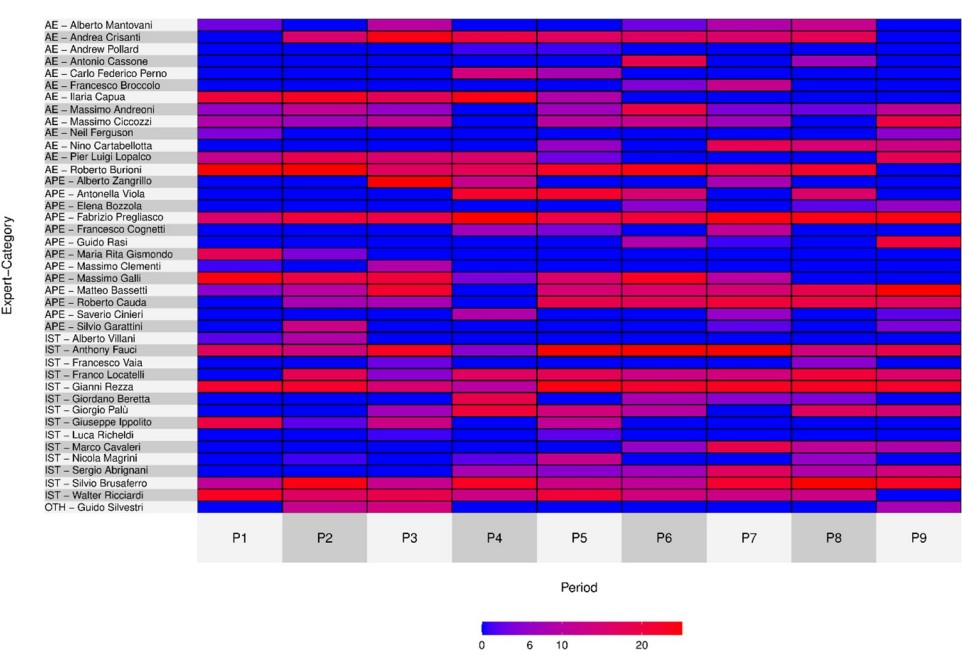

**Fig 2. Heatmap share for experts across time, in the 24 months' time span, January 1st, 2020 –December 31st, 2021 (B2 COVID EXPERT SUBSET only articles related to technoscience, N = 11,940).**

task force about the pandemic, never lost his prominence across time. Although some oscillation is visible for experts in both heatmaps, the general trend shows a steady leading role for firstcomer scientists as experts among the Italian media scientists. Hence, continuity is a feature characterising scientist publicly dealing with the Sars-Cov-2 pandemic. This resonates quite well with the literature on experts' media presence and also with what we have previously observed comparing the media visibility of the top 25 scientists before the pandemic.

Besides continuity, we record some interesting exceptions. Some experts, such as Lopalco and Capua—both academic experts (AE)—lost their initial prominence, fading out of the top tier of most prominent scientists. They did not disappear actually; rather they just lose media visibility temporarily, and this may hint to a different process of media sieving. Hence, becoming a media expert is not the outcome of an irreversible process; the complexity of media sieving entails the possibility of laying provisionally in a latent condition and then to gain the media scene again rather than a fully irreversible expulsion from the media. This the case for Antonella Viola, whom we classified as an academic and personal expert (APE), who figured low in early periods (P1 and P2) while raised in the summer 2020 (P3 and P4) and then faded away, sliding back down in media visibility; then she had a certain resurgence in P7 when she received anonymous intimidation. Viola's case is illustrative since it represents a specific, possible, not necessary residual pattern in a media expert's career; alongside the scientific credentials, prominence may be the output of a combination of elements, including what counts as relevant news for newspapers. This is also the case of Lopalco, who after having decreased his visibility in P7 e P8 regained the mediatic scene in P9 because he resigned polemically from assessor of the government of Puglia region. Comparing the two heatmaps, we recorded the same trends, with very limited variation in terms of intensity in prominence and continuity, as in the case of Bassetti who is more constantly in the top tier if we consider the corpus B1 (Fig 1) compared to the corpus of articles related to technoscience (Corpus B2, Fig 2). As it concerns the experts' categories, by comparing the two heatmaps we note immediately in both cases that institutional experts (IST) are also largely consistent in their capability to stay on the frontstage during all the periods, even if there is some decrease in visibility. This is also the case of some AEs and APEs, such as Crisanti and Burioni or Galli and Bassetti, even if institutional experts (IST) hold the media scene with less discontinuities. In any case, given that there are no statistical differences between trends for different kind of experts, the similarities and the differences we have reported cannot be directly connected to the category; media careers for visible experts may follow other factors, as suggested by our evolutionary model.

Beyond the three factors already outlined—i.e. occupying an institutional position, having a previous media visibility, and matching academic credentials with media capability to offer explanations and practical suggestions deemed useful by the media—a closer inspection of data concerning newspapers can be useful for identifying others factors that act as *media sieves*.

The first is related to the mediatic evolution of the issue, which has been observed to change its frames or "interpretative packages" [66] according to different phases: whether we use a two-step [67] or a three-step model [68], we can agree that hot crises in media begins with the sounding of the alarm and then turn to a more reassuring register.

In the Covid-19 pandemic, media seem to be inclined to choose experts' profiles according to the different ways of framing the outbreak. Nonetheless, some expert categories—as shown in Figs 1 and 2 —tend to maintain prominence in the media scene more than others. Thus, while 'institutional experts' (IST) tend to maintain high visibility during all periods, some AEs and APEs present greater or lesser visibility depending on the specific framing of the pandemic that sets the 'problem of the moment'. The problem of the moment is reified through questions such as 'Is the virus lethal?', 'How can we avoid contagion?', 'How will the pandemic evolve?',

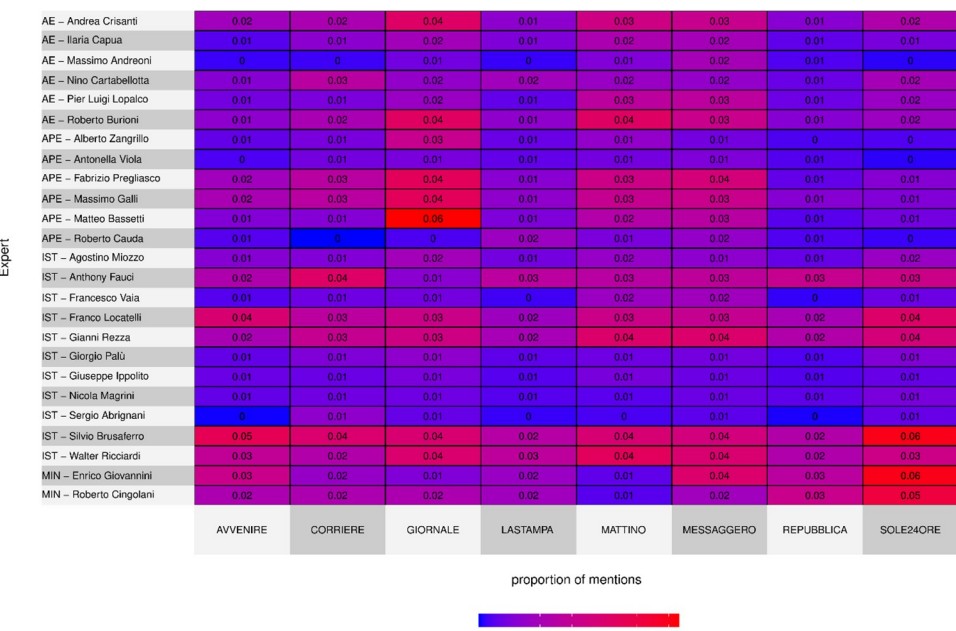

**Fig 3. Heatmap showing experts' presence across the newspapers monitored in the 24 months' time span, January 1st, 2020 –December 31st, 2021 (B2 COVID EXPERT SUBSET N = 25,550).**

'Are the measures adopted by the government adequate?', which experts engaged on media are then asked to answer.

Exactly as postulated by Nowotny [39], those questions are defined by sources external to the scientific community, such as for instance the media agenda [69], often in close collaboration with the political agenda [12]. Here again we can observe that becoming a *media star* depends on being suitable for an additional media criterion, i.e. being the right expert at the right moment.

Another factor acting as a *media sieve* regards the correspondence between what experts say and the editorial line of the specific newspaper.

As can be seen in Fig 3, the presence of the top 25 experts has a weight that varies according to the newspaper, with the only exception being Silvio Brusaferro—i.e. the President of the Higher Institute of Health, and spokesperson from 17 March 2021 onwards of the Governmental Scientific Technical Committee—who is frequently cited by all eight newspapers examined.

Each newspaper has some reference experts; we can suggest that some scientists become *media stars* according to their suitability for a specific venue. This identifies another element in support of the MEEM as an interpretative tool to understand how a scientist can assume the role of *media star*.

## 5. Concluding remarks

Through the analysis proposed in this paper we gave account of MEEM: an evolutionary model to interpret trajectories of scientists from entering the media arena up to becoming a *media star*.

In order to interpret those trajectories, we build upon a review of scientists as experts in the media; we then distilled the main features of experts as they appear on the news and the role of expert itself. In a context of advanced mediatisation of science [25] we can distinguish a kind

of push-and-pull process: on the one hand, scholars have described how scientists actively push to get access to the media sphere and keep media attention on them [28, 29]; on the other hand, media pull scientists into the media arena. Media, indeed, are eager to have scientists explaining complex issues not only because technoscience is newsworthy, but also because several of our contemporary debates and controversies are technoscientific ones [19, 20, 22, 23]. Being a "visible scientist" [31] has pros and cons, as described by researchers in the PCST debate [1], but the literature we surveyed lacks a model able to interpret the career of scientists as media experts on their trajectory to becoming *media star*s.

Informed by the above-mentioned aspects, we proposed an attempt to fill this gap in the literature. We made use of the pandemic as a period of scientists' over-exposure; as already shown elsewhere [15], the share of technoscientific content in the daily press has grown as never before in the Italian context during the pandemic and this represented a key opportunity for investigating scientists in the media. The pandemic context allowed us to magnify some features already recorded in the literature, making them easier to be observed through media.

We concentrated on the credentials that scientists need and on the sieving processes enacted by newspapers; the combination of these two defines the MEEM. Regarding the credentials, we recorded as particularly relevant i) institutional role/position, ii) previous media visibility, and iii) match between scientific credentials and media competence. In our analysis, we further demonstrated that the media presence is not necessarily related with scientific credentials, as the low correlation between h-index and media presence has shown. Thus, we called our model an evolutionary one since only those scientists that have some features best adapted to a media environment can *survive* in the media arena.

Moreover, we noticed that being a *media star* is not an irreversible achievement. Indeed, we recorded in some cases a sort of up-and-down trajectory that put scientists initially on the front-stage, then send them back under the threshold of media visibility, and then bring them again within the top positions; although they possessed those key credentials that allowed media to enrol them, they only maintained a high presence in the media for a limited time span.

This fact drove us to consider the active role of media—newspapers in our case—as sieves. One factor here is the general framing of the pandemic: not all the types of expertise have the same value in the same moment, precisely because hot crises in media alternate between the sounding of the alarm and the phase of reassurance. Secondly, some newspapers may prefer specific experts as media actors according to their ability to align to editorial choices.

In short, we collected evidence that celebrity for scientists in newspapers can be seen as evolutionary in the sense of being adapt to a specific environment such as the media arena.

Admittedly our modelling attempt is limited because of the pandemic context, in which we probably took advantage of an acceleration of the events: the outbreaks pushed governments to contain the contagion and the emergency dominated the coverage in Italy and everywhere else. Therefore, we noticed an increase in the presence of scientists in the daily quality press. The salience of technoscientific news coverage grew from 8% to 10% on average; this specific condition helped in our analysis, but nonetheless it also calls for our MEEM to be tested further in the future, both on other corpora reporting in regular times and in other cultural contexts

Furthermore, although newspapers can be regarded as a proxy of what happens in the media sphere providing significant indications on the media presence of scientists, the MEEM model needs to be tested further with data concerning other media, such as television and social media. Nevertheless, it might be regarded as a useful benchmark—not necessarily related only to pandemic periods–for scholars working into the field of journalist studies, public understanding of science (PUS), public communication of science and technology (PCST) and more broadly media studies.

Lastly, even if the previous considerations highlight the analytical limits of the model, it could be of interest anyway for a better management of the relations between the media and scientists in the context of the PCST.

## Supporting information

**S1 File. TIPS project named entity resolution and entity linking procedure for counting scientists mentions in newspaper articles.**
(DOCX)

## Author Contributions

**Conceptualization:** Federico Neresini, Paolo Giardullo, Barbara Morsello.

**Data curation:** Paolo Giardullo, Emanuele Di Buccio, Alberto Cammozzo, Andrea Sciandra.

**Formal analysis:** Emanuele Di Buccio, Andrea Sciandra.

**Methodology:** Federico Neresini, Emanuele Di Buccio, Alberto Cammozzo, Andrea Sciandra.

**Supervision:** Federico Neresini.

**Writing – original draft:** Federico Neresini, Paolo Giardullo, Emanuele Di Buccio, Barbara Morsello, Alberto Cammozzo, Marco Boscolo.

**Writing – review & editing:** Federico Neresini, Paolo Giardullo, Barbara Morsello, Andrea Sciandra.

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
