## [Decision Letter · Decision Letter 0]

24 Feb 2023

PONE-D-22-31059WHEN SCIENTIFIC EXPERTS COME TO BE MEDIA STARS: AN EVOLUTIONARY MODEL TESTED BY ANALYSING CORONAVIRUS MEDIA COVERAGE ACROSS ITALIAN NEWSPAPERSPLOS ONE

Dear Dr. Neresini,

Thank you for submitting your manuscript to PLOS ONE. After careful consideration, we feel that it has merit but does not fully meet PLOS ONE’s publication criteria as it currently stands. Therefore, we invite you to submit a revised version of the manuscript that addresses the points raised during the review process.

We look forward to receiving your revised manuscript.

Kind regards,

Ramona Bongelli, Ph.D.

Academic Editor

PLOS ONE

Journal Requirements:

3. Please include your tables as part of your main manuscript and remove the individual files. Please note that supplementary tables (should remain/ be uploaded) as separate "supporting information" files.

Reviewers' comments:

Reviewer's Responses to Questions

**Comments to the Author**

1. Is the manuscript technically sound, and do the data support the conclusions?

Reviewer #1: Yes

Reviewer #2: Yes

2. Has the statistical analysis been performed appropriately and rigorously? 

Reviewer #1: N/A

Reviewer #2: N/A

3. Have the authors made all data underlying the findings in their manuscript fully available?

Reviewer #1: No

Reviewer #2: Yes

4. Is the manuscript presented in an intelligible fashion and written in standard English?

Reviewer #1: Yes

Reviewer #2: Yes

5. Review Comments to the Author

Reviewer #1: Thank you for the opportunity to read this interesting article, which deals with a topical issue and may attract the attention of scholars dealing with media communication from different disciplinary perspectives.

The authors present an original study, mainly aimed at analysing several processes through which some scientists can become media experts and even “media stars”, specifically investigating the Italian context during the Covid-19 pandemic emergency, when the role of the public expert became crucial, and some scientists became ‘visible’ in the media, or even over-exposed

They propose an analytical model (MEEM) that they define as “evolutionary” (in a Darwinian sense), since it attempts to explain how and according to which characteristics some scientists can emerge, adapt and remain for a long time present in the media environment to which they do not originally belong.

The manuscript is well-organised and clearly written. The study appears to be correctly placed in the context of the relevant literature. Data and analysis support the authors’ claims. Conclusions are presented in a quite appropriate fashion and supported by the data.

From a methodological point of view, the article presents a coherent descriptive quantitative analysis, clearly presented and accompanied by tables and figures useful for understanding the analysis carried out and the results obtained. Therefore, I have no particular remarks to make regarding these sections.

I will limit myself to the following comments/suggestions, hoping they may be useful.

- Please consider reviewing the abstract, which sounds a bit repetitive in some of its parts

- Line 42. Consider starting a new paragraph

- Line 340: Regarding the nine periods into which the pandemic emergency in Italy has been divided, is it possible to cite an official source (perhaps through a link or website) or have the authors simply and independently relied on the dates of the Decrees-Laws promulgated by the Italian government?

- In the Introduction or Concluding remarks, it might be useful to identify the potential target readership/scholars interested in this study, since no explicit reference is made to disciplinary fields or research macro-areas.

- Finally, the main minor point I would like to make concerns a factor that the authors did not explicitly consider and which, in my opinion, they could instead discuss (or at least mention) in the Concluding remarks. The study identifies the “media stars” from the analysis conducted on corpora of articles from the eight most important Italian newspapers, but today the main source of visibility and influence is represented more by TV and social media than by traditional newspapers, even in their online versions. Therefore, I feel it is important to emphasise, not only the limitations of this analysis, but also the possible relationships between visibility in daily newspapers and in other media with greater and more widespread use. In short, I think that it could be at least discuss, among the characteristics that this study could not take into account, the possible impact of the experts’ personal media exposure (e.g. whether they are regular guests on TV programmes, whether they have active social media pages, blogs, etc.).

Reviewer #2: The paper deals with very interesting, actual and original issues. I read it with a great interest the argumentation concerning the relationship between media visibility and scientific reputation: a wide problematization of the ambivalence between scientific credibility and media requirement has been have been proposed.

In addition, research questions and procedures are well explained and organized; an extensive amount of data is under analysis.

However, I propose just some observations that, I believe, can improve the paper:

- pag. 23: As for Lopalco, as you probably know, at the end of 2020 he had a political charge as Assessor for Welfare in Puglia. Maybe this factor also influenced his "losings" in media; on the other side, this information could take a reflection about how being a "star" can help to gain new opportunities in other life domains

- you analysed data coming from newspapers. Do you think the situation could me similar or different when compared with tv?

- limitation, strength and practical implications could be better argued.

6. PLOS authors have the option to publish the peer review history of their article (what does this mean?). If published, this will include your full peer review and any attached files.

Reviewer #1: No

Reviewer #2: No

---

## [Author Response · Author response to Decision Letter 0]

6 Apr 2023

♦ As required by both rev.#1 and rev.#2, the possible relationships between visibility in daily newspapers and in TV or social media has been discussed both within the 4.1 session and in the concluding remarks where limits and significance of testing the proposed MEEM model only in newspapers are discussed. It has been also clarified that the research did not collect data regarding TV and/or social media, but it is possible to argue that what is observed in the newspapers can also offers indications about the presence of scientists in other media. 

♦ As required by rev.#1 and rev.#2, some hints about disciplinary fields and potential readership/scholars have been added in the conclusion.

♦ About other requests by rev.#1 :

- Please consider reviewing the abstract, which sounds a bit repetitive in some of its parts �

the abstract has been reviewed, also deleting redundant passages;

- Line 42: Consider starting a new paragraph � done;

- Line 340: Regarding the nine periods into which the pandemic emergency in Italy has been divided, is it possible to cite an official source (perhaps through a link or website) or have the authors simply and independently relied on the dates of the Decrees-Laws promulgated by the Italian government? � it has been added the link to the Italian Ministry of Health where all the official documents, decrees and rules are listed.

♦ About other requests by rev.#2 :

- pag. 23: As for Lopalco, as you probably know, at the end of 2020 he had a political charge as Assessor for Welfare in Puglia. Maybe this factor also influenced his "losings" in media; on the other side, this information could take a reflection about how being a "star" can help to gain new opportunities in other life domains � a brief comment concerning the media trajectory of Lopalco has been inserted in 4.2 session.

---

## [Editor Report · Decision Letter 1]

11 Apr 2023

WHEN SCIENTIFIC EXPERTS COME TO BE MEDIA STARS: AN EVOLUTIONARY MODEL TESTED BY ANALYSING CORONAVIRUS MEDIA COVERAGE ACROSS ITALIAN NEWSPAPERS

PONE-D-22-31059R1

Dear Dr. Neresini,

We’re pleased to inform you that your manuscript has been judged scientifically suitable for publication and will be formally accepted for publication once it meets all outstanding technical requirements.

Kind regards,

Ramona Bongelli, Ph.D.

Academic Editor

PLOS ONE

Additional Editor Comments (optional):

Thank you for revising the paper according to the reviewers' instructions. The paper can be accepted as it stands, but I would ask you to (a) check the correct functioning of the doi you have indicated for the corpus and (b) specify which eight newspapers are cited and whether you have consulted their online version as you seem to understand from the TIPS website.

(a) The doi indicated in the submission 10.5281/zenodo.7712714

appears not to work. Make sure it is correct and that will be available after publication.

(b) Line 308: reference is made to the eight most important Italian newspaper. I think it is important to list them. There is no trace of what these 8 newspapers are

Also from the site cited in the publication (11) Techno-Scientific Issues in the Public Sphere (TIPS). EASST Review, accessible at this link https://www.easst.net/article/techno-scientific-issues-in-the-public-sphere-tips/ one can only read "eight most important Italian newspapers". Among other things, the very first lines describing the project you can read that “The TIPS project is based on the idea of using mass media and online newspapers, in particular”. Your article does not mention online newspapers. If this is the case, I think it is better to specify.
---

## [Editor Report · Acceptance letter]

17 Apr 2023

PONE-D-22-31059R1 

When scientific experts come to be media stars: an evolutionary model tested by analysing coronavirus media coverage across Italian newspapers 

Dear Dr. Neresini:

I'm pleased to inform you that your manuscript has been deemed suitable for publication in PLOS ONE. Congratulations! Your manuscript is now with our production department. 

Kind regards, 

on behalf of

Professor Ramona Bongelli 

Academic Editor

PLOS ONE